# Application Value of Antimicrobial Peptides in Gastrointestinal Tumors

**DOI:** 10.3390/ijms242316718

**Published:** 2023-11-24

**Authors:** Qi Liu, Lei Wang, Dongxia He, Yuewei Wu, Xian Liu, Yahan Yang, Zhizhi Chen, Zhan Dong, Ying Luo, Yuzhu Song

**Affiliations:** 1College of Life Science and Technology, Kunming University of Science and Technology, Kunming 650500, China; 2Medical College, Kunming University of Science and Technology, Kunming 650500, China

**Keywords:** antimicrobial peptides, gastrointestinal cancer, anticancer mechanism

## Abstract

Gastrointestinal cancer is a common clinical malignant tumor disease that seriously endangers human health and lacks effective treatment methods. As part of the innate immune defense of many organisms, antimicrobial peptides not only have broad-spectrum antibacterial activity but also can specifically kill tumor cells. The positive charge of antimicrobial peptides under neutral conditions determines their high selectivity to tumor cells. In addition, antimicrobial peptides also have unique anticancer mechanisms, such as inducing apoptosis, autophagy, cell cycle arrest, membrane destruction, and inhibition of metastasis, which highlights the low drug resistance and high specificity of antimicrobial peptides. In this review, we summarize the related studies on antimicrobial peptides in the treatment of digestive tract tumors, mainly oral cancer, esophageal cancer, gastric cancer, liver cancer, pancreatic cancer, and colorectal cancer. This paper describes the therapeutic advantages of antimicrobial peptides due to their unique anticancer mechanisms. The length, net charge, and secondary structure of antimicrobial peptides can be modified by design or modification to further enhance their anticancer effects. In summary, as an emerging cancer treatment drug, antimicrobial peptides need to be further studied to realize their application in gastrointestinal cancer diseases.

## 1. Introduction

Antimicrobial peptides (AMPs) are short amino acid sequences found in bacteria and mammals, typically containing 12 to 50 L-amino acids, with a net positive charge of +2 to +9 at neutral pH [1,2,3]. According to the biosynthetic pathway, AMPs can be divided into two categories: ribosomal synthesis and non-ribosomal synthesis. Post-translational modifications may occur in many AMPs synthesized by ribosomes, resulting in amino acids with nonprotein structures, such as Nisin [4]. Non-ribosomal synthesis involves non-ribosomal peptide synthetases (NRPSs), which are mainly found in bacteria and fungi, such as bacitracin [5]. AMPs are also released from immune cells and epithelial cells in different human organs. Most AMPs share common characteristics, including hydrophobicity, cationic properties, and amphiphilic structures, which determine their broad-spectrum antimicrobial activities against bacteria, fungi, protozoa, and viruses [6,7,8]. This unique molecular structure and antimicrobial activity make AMPs promising as an alternative to antibiotics and widely studied, as antibiotic resistance is currently a major challenge for the effective treatment of bacterial infections. In addition, AMPs have also shown anticancer activity [9,10]. Many AMPs, known as anticancer peptides (ACPs), can destroy the structure of tumor cells or inhibit the proliferation and metastasis of tumor cells and cause little damage to normal cells [11,12]. Compared with current anticancer strategies, AMPs have a lower likelihood of developing resistance during treatment and produce less harmful effects on normal cells [13,14].

Cancer, a tumor or malignancy, is the leading cause of death, affecting nearly 10 million people, and the second leading cause of death in developing countries [15,16]. According to data from the World Health Organization (WHO), the mortality from gastrointestinal tumors accounted for about 35% of all malignant tumor mortality in 2020, mainly including oral cancer, esophageal cancer, gastric cancer, liver cancer, pancreatic cancer, and colorectal cancer, which seriously endanger human health [17]. Surgical resection, radiotherapy, chemotherapy, immunotherapy, and antibody-based molecules are common cancer treatments [18,19,20,21]. These approaches all face challenges and limitations in the field of gastrointestinal cancer treatment. The cure rate is relatively low, and these treatments can affect solid cells, leading to a series of adverse reactions such as severe nausea and vomiting, alopecia, and cardiac toxicity [22,23,24]. With these concerns in mind, researchers and cancer patients are hoping to reduce the burden of cancer with a more specific treatment that has fewer side effects and a lower rate of cancer recurrence [25,26,27].

In recent years, more and more evidence has shown that the high selectivity and low drug resistance of AMPs can effectively inhibit the metastasis and proliferation of cancer cells, which provides a new strategy for cancer treatment [28,29,30]. According to the current study, the use of AMPs in gastrointestinal tumors is an effective approach for the development of novel anticancer drugs (Figure 1). This article reviews the application of AMPs in oral cancer, esophageal cancer, gastric cancer, liver cancer, pancreatic cancer, and colorectal cancer, and provides a new vision and ideas for the development and clinical application of new drugs for digestive tract tumors.

## 2. Antimicrobial Peptides against Gastrointestinal Tumors

Gastrointestinal tumors are tumors that grow on the digestive system, including oral cancer, esophageal cancer, gastric cancer, liver cancer, pancreatic cancer, and colorectal cancer. Recent studies have found that AMPs have anticancer activity against a variety of digestive tract tumor cells. Among them, LL-37, as the only human member of the antimicrobial peptide family, has shown potent anticancer effects, showing antitumor activity against colorectal cancer, gastric cancer, and liver cancer cells [31]. CopA3 has antitumor effects on colorectal cancer, gastric cancer, and pancreatic cancer cells, and cecropin series AMPs showed antitumor effects on gastric cancer, liver cancer, and esophageal cancer cells. In addition, the research progress of various AMPs in gastrointestinal tumors is introduced in the following section.

### 2.1. Oral Cancer

Oral squamous cell carcinoma (OSCC) is the most common oral cancer, with poor prognosis and high mortality. There are about 300,000 diagnosed cases and 150,000 deaths worldwide every year [32,33,34]. The main treatment for oral squamous cell carcinoma is surgical resection, but the prognosis of survival is not high [35,36,37]. Commonly used first-line chemotherapy drugs include cisplatin, carboplatin, 5-fluorouracil, etc. However, these drugs not only kill cancer cells but also damage normal healthy cells [38,39,40,41]. Considering these factors, it is necessary to find new therapeutic methods for oral squamous cell carcinoma, and some AMPs have emerged as potential drugs for the treatment of oral squamous cell carcinoma (Table 1).

As early as 2004, it was reported that hCAP18 (109–135), an analog of LL-37, induced apoptosis in SAS-H1 cells through a caspase-independent pathway [46]. KI-21-3, a shortened fragment of LL-37, has obvious oncolytic properties in SCC-4 cells through antiproliferation and caspase-3 apoptotic pathways [45]. Human phthalamide antimicrobial peptide (CAMP) and LL-37 C-terminal deletion mutant (CDEL) were also shown to induce apoptosis of HSC-3 cells through the P53-Bcl-2/BAX signaling pathway [44]. These results all suggest that LL37 and its analogs have varying degrees of influence on the development of oral squamous cell carcinoma and may act as tumor suppressors in oral squamous cells.

Human β-defensin (HBD) is produced by the epithelial cells of many organs. Among the numerous types of HBD, HBD-1, HBD-2, and HBD-3 have been well studied. Among them, HBD-1 could inhibit the proliferation of oral squamous cell carcinoma BHY cells, but BHY cells increased after the stimulation of HBD-2 and -3 [49]. In a further study, Qi Han et al. found that exogenous expression of HBD-1 inhibited the migration and invasion of oral squamous cell carcinoma lines; however, the specific mechanism remains unclear [50]. For HBD-2, Yoshitaka Kamino et al. found that increased HBD-2 expression inhibited SAS cell proliferation and invasion [52].

### 2.2. Esophageal Cancer

Esophageal cancer is the sixth most common cancer worldwide, with poor prognosis and a low overall survival rate [53]. At present, there are few studies on AMPs in esophageal cancer, mainly several types of cecropin, such as cecropin A, cecropin B, cecropin D, and cecropinXJ (Table 2). Cecropin A, cecropin B, and cecropin D can exert anti-esophageal cancer activities through the mitochondrial apoptosis pathway [54,55,56]. CecropinXJ was found to induce cytoskeletal disruption such as microtubule depolymerization and actin polymerization, as well as to regulate the expression of cytoskeletal protein genes, resulting in cytotoxicity against esophageal cancer Eca109 cells [57]. In addition, Shangjie Liu et al. found that LvHemB1 could also be selectively toxic to esophageal cancer through the mitochondrial apoptosis pathway, and EC190 cell viability decreased by 49.1% after treatment with 50 µg/mL for 24 h, with no significant effect on the proliferation of noncancer cell lines [58]. Compared with cecropin, LvHemB1 has a higher toxicity to EC190 cells.

### 2.3. Gastric Cancer

Gastric cancer is the fifth most common cancer in the world, accounting for 7.7% of all cancer deaths [59]. Because of the few symptoms caused by the early stage, gastric cancer is usually not diagnosed in time, and metastasis occurs in 80% to 90% of patients with gastric cancer [60,61]. Despite improvements in diagnosis and treatment, the overall survival of patients with gastric cancer is <40% [62]. At present, more than ten AMPs have been studied for the treatment of gastric cancer (Table 3).

Melittin, an antimicrobial peptide derived from bee venom, has been extensively studied on a variety of cancer cells. Amir Mahmoodzadeh et al. first reported the toxicity of melittin isolated from Iranian bee venom to gastric cancer AGS cells. Even at very low concentrations (0.5 mg/mL) of melittin treatment (6–24 h), melittin inhibited the proliferation of AGS cells. At the concentration of 1 mg/mL, loose integrity of the cell membrane was observed, which was a marker of cell necrosis and death [74]. Caroline Soliman et al. also studied the transient effect of melittin on gastric cancer AGS cells (within 15 min). They found that swelling, membrane blebbing, and rupture of cells occurred within a few seconds after high-dose melittin treatment and complete cell death occurred within 15 min [70]. The cause of death in most cancer patients is directly related to recurrence and cancer cell metastasis. In the study by Jye-Yu Huang et al., it was shown that melittin can reduce the expression of related proteins and inhibit AGS cell migration and invasion through multiple pathways (Table 3), which indicates that melittin has the potential to treat metastatic gastric cancer [65]. In addition, melittin has been found to induce apoptosis in human gastric cancer cells SGC-7901 by activating the mitochondrial pathway, which is a further understanding of the anticancer mechanism of melittin [61]. However, in the above studies, melittin showed not only anticancer activity but also strong hemolytic activity, which is the biggest challenge in developing melittin as a therapeutic agent for gastric cancer.

### 2.4. Liver Cancer

Liver cancer is the fourth most common cause of cancer-related death worldwide, and the most common form is hepatocellular carcinoma, with a 5-year survival rate of approximately 18% [76,77]. Sorafenib has been used as a new targeted drug for the treatment of liver cancer since 2007. It is the only systemic treatment drug approved by the FDA for the treatment of advanced unresectable hepatocellular carcinoma [78]. However, it has high toxicity and drug resistance, which may affect the function of normal cells and cause some adverse reactions common to antiangiogenic drugs [79]. Due to the high degree of selective toxicity of AMPs, more than 20 kinds of AMPs derived from humans, insects, animals, and plants and via artificial synthesis have been widely used in the study of liver cancer treatment (Table 4).

Cecropin is an antimicrobial peptide from *Musca domestica*. Xiaobao Jin et al. found that cecropin could inhibit the proliferation of human liver cancer BEL-7402 cells in a dose- and time-dependent manner through the extrinsic apoptotic pathway [83]. Further studies showed that cecropin also showed inhibitory potential for liver cancer cell metastasis, and inhibited the adhesion and migration of human liver cancer BEL-7402 cells [82]. Purified cecropin-B also had anti-HCC activity with a semi-inhibitory concentration of 25 µg/mL on HepG2 cells, which was safe for human normal lung WI-38 cells with a cytotoxicity of 0.92% [86]. Cecropin XJ shares 98% of its identity with cecropin B [106]. Lijie Xia et al. found that cecropin XJ could inhibit the proliferation and induce apoptosis of Huh7 cells in vitro through the mitochondrial apoptosis pathway [85]. A cecropin B analogue, cecropin-p17, was also found to exert anti-HCC activity both in vitro and in vivo, which may be related to cell apoptosis [99]. It has been shown that the bacteriocin Nisin can also play a role in HepG2 cell apoptosis through the mitochondrial pathway, and more importantly, Nisin treatment can lead to a reduction in the expression of the EMT transcription factor TWIST1, which can misregulate the sensitivity to drug treatment [96,103].

### 2.5. Pancreatic Cancer

Pancreatic cancer is a common cause of cancer death worldwide, with a mortality rate of about 4.5% in both men and women [17]. The antimicrobial peptide CopA3, derived from dung beetle defensins, was found to dose-dependently inhibit the growth of human pancreatic cancer MIA-PaCa2 with an IC50 of 61.7 µM [105]. This is the first study on the application of AMPs in pancreatic cancer. CopA3 has potential application in the treatment of pancreatic cancer, but its antitumor mechanism still needs to be further elucidated.

### 2.6. Colorectal Cancer

Colorectal cancer (CRC) is the third most common type of cancer worldwide, with high morbidity and mortality [107]. Oxaliplatin and fluorouracil are common chemotherapeutic drugs, but oxaliplatin can cause severe peripheral nerve injury, and fluorouracil can also cause adverse gastrointestinal reactions and liver injury [108]. In recent years, dozens of AMPs have been widely used in the treatment of colorectal cancer because of their high specificity and low occurrence of side effects. They have different degrees of cancer-killing effects in various colon cancer cell lines (Table 5).

The antimicrobial peptide LL37 and its residues and analogs present in the human body have been widely studied in colon cancer. Shun X. Ren et al. reported that antimicrobial peptide LL37, which exists in the human body, induces caspase-independent apoptosis by upregulating Bax and Bak and downregulating Bcl-2, leading to nuclear translocation of AIF and EndoG to induce caspase-independent apoptosis and thus inhibiting the occurrence of colon cancer [123]. FK-16, as a 17–32 residue of LL37, also induced AIF-dependent/EndoG-dependent apoptosis and autophagic cell death through the p53-Bax/Bcl-2 cascade commonly observed in colon cancer cells [113,123]. Compared with LL37, FK-16 has a more significant effect on the activity of colon cancer cells, showing better anticancer activity, and the shortened length of FK-16 can also reduce the production cost associated with peptide synthesis. FF/CAP18 is an analog of LL-37; Kengo Kuroda et al. have found that 10 μg/mL FF/CAP18 can induce partial mitochondrial membrane depolarization at the early stage of apoptosis, and high-dose treatment (40 μg/mL) can lead to late apoptosis. Glycolysis and the tricarboxylic acid cycle are inhibited to reduce ATP production, resulting in the absence of most metabolites [140]. As a mimetic of LL-37, CSA-13 could induce cell cycle arrest and inhibit the proliferation of HCT116 cells [130].

In addition, Br-J-I, a halogen-derived antimicrobial peptide isolated from royal honeybee jelly, did not induce cell death by apoptosis or membrane destruction. It had little cytotoxicity against colon cancer cells but showed antibacterial activity against *Fusobacterium nucleorum* (*Fn*) [127]. Some studies have reported that *Fn* is closely related to the occurrence and development of CRC [141,142,143,144]. Therefore, Br-J-I can directly kill *Fn* to indirectly inhibit colorectal cancer growth [127].

## 3. Anticancer Mechanisms of Antimicrobial Peptides against Gastrointestinal Tumors

In recent years, AMPs have become a research hotspot for antitumor drugs, and their antitumor mechanisms have been reported in experimental studies of digestive tract tumors. Most AMPs interact with membranes and form special pore channels, possibly through barrel stave-, carpet-, and detergent-“like” mechanisms, that allow AMPs, ions, or other substances to reach intracellular targets, thereby triggering a variety of antitumor mechanisms [145,146], including induction of apoptosis, autophagy, disruption of the cell membrane, arrest of the cell cycle, inhibition of metastasis, and disruption of the cytoskeleton. Among them, the mechanism of cytoskeleton disruption has not been extensively studied. This article explains the following five important anticancer mechanisms of AMPs (Figure 2).

### 3.1. Cell Apoptosis

Apoptosis is a type Ⅰ programmed cell death process. Two major apoptotic pathways exist, the first being the extrinsic pathway (death receptor pathway), triggered by the CD95 (Fas) death receptor and some members of the tumor necrosis factor-α (TNF-α) receptor superfamily [128]. The second is the endogenous pathway (mitochondrial-mediated pathway), which is further divided into caspase-dependent and caspase-independent pathways. Some studies have found that AMPs can cause mitochondrial dysfunction to exert anticancer activity. Due to the negative charge of mitochondria, AMPs may target mitochondria, destroy the integrity of mitochondrial membranes, or control the permeability of mitochondrial membranes by regulating the ratio of Bax/Bcl-2 and the release of ROS [147,148,149]. This eventually leads to the release of cytochrome C into the cytoplasm, which binds to APAF-1 and caspase-9 to form apoptotic bodies, activates downstream caspase-3, and eventually leads to caspase-dependent apoptosis [147,150]. In addition, when cells are stimulated by internal apoptotic factors, the proapoptotic protein AIF present in the mitochondria is transferred to the nucleus, leading to DNA damage and causing caspase-independent apoptosis. Among them, LL37 and FK-16 induced caspase-independent apoptosis of colon cancer cells by inducing the nuclear translocation of AIF and EndoG through the upregulation of Bax and Bak and the downregulation of Bcl-2 [113,123].

### 3.2. Autophagy

In the study of AMPs in the treatment of tumors, it is found that there may be an interaction between autophagy and apoptosis. Autophagy is a type Ⅱ programmed death process, which includes four key steps: initiation, nucleation, maturation, and degradation [151]. The formation of autophagy leads to the enzymatic conversion of LC3-Ⅰ to membrane type LC3 (LC3-Ⅱ). As an autophagy marker, LC3-II participates in the formation of autophagosome membranes [152,153]. Beclin 1 plays an important role in autophagy and tumorigenesis. It can mediate the localization of autophagy proteins to phagosomes and induce the formation and maturation of autophagosomes [154,155]. Normally, the expression of beclin 1 is increased during autophagy. LFcinB 25, GW-H1, and bovine lactoferricin B increased LC3-II and beclin-1 at the same time in the early stage of treatment, LC3-II began to decrease in the later stage, beclin-1 increased continuously, and autophagy was inhibited [64,67]. Shun X. Ren et al. found that elimination of autophagy can make FK-16 promote apoptosis; FK-16 activates p53 to upregulate Bax and downregulate Bcl-2 to induce apoptosis; Bcl-2 and Bcl-xL seem to be important factors in autophagy and inhibit autophagy by binding to beclin-1 [113]. KT2 inhibited autophagy by reducing the expression of LC3-I, Atg5, Atg7, Atg16L1, and beclin-1 in cells, but the effect between autophagy and apoptosis was not further studied [120]. The lysosome is an important regulator of autophagy. Jiali Zeng et al. found that M1-8 can colocalize with lysosomes, leading to lysosomal rupture, release cathepsin D (m-CTSD) into the cytoplasm, activate caspases and change mitochondrial membrane potential, and finally induce cell apoptosis [95]. Smp43 and Smp24, two AMPs derived from scorpion venom, increased the expression of autophagosome formation marker LC3A/B-Ⅱ in a dose-dependent manner, and increased autophagy by regulating the PI3K/Akt/mTOR signaling pathway [88,97].

### 3.3. Cell Cycle Arrest

Some of the currently reported AMPs can inhibit cancer cell proliferation through cell cycle arrest, which includes interphase G1, S, G2, and mitotic M phases. This process is tightly regulated by cyclin and cyclin-dependent kinases (CDKS). When cells in the G0 phase are stimulated, they express cyclin C, cyclin D, and cyclin E. Cyclin D binds to a variety of kinases, mainly CDK4, and cyclin E binds to a variety of kinases, mainly CDK2, and cells enter the S phase and begin DNA synthesis [156,157]. The S and M phases are mainly regulated by cyclin A and cyclin B. Debasish Kumar Dey et al. found that CopA3 treatment inhibited the expression of cyclin and CDK and arrested the G1 phase of the cell cycle [126]. Cyclin E1 is involved in the regulation of G1/S transition, and when its expression is inhibited, it can effectively promote S phase progression. C. Freiburghaus et al. found that lactoferrin inhibited cyclin E1 expression and prolonged the S phase of cancer cells [134]. Cell cycle arrest prevents cells from undergoing mitosis, leading to accumulation of DNA, and DNA content reflects cell cycle arrest. As measured using cell cycle DNA content, both CSA-13 and the bacteriocin enterocin-A arrested cancer cells in the G1 phase, and overexpression of DEFA5 inhibited the G1/S phase of the cell cycle [116,130]. P53 regulates the expression of a series of genes involved in the G1/S and G2/M transitions [158]. Pardaxin may lead to G2/M phase-induced cell arrest through the regulation of p53 and cyclin B1 [42].

### 3.4. Membrane Destruction

Many AMPs are cationic amphiphilic peptides that can bind to negatively charged cell membranes through electrostatic interactions, leading to membrane disruption. The surface of cancer cells is negatively charged because the outer membrane of cancer cells expresses anionic components such as glycoproteins and phosphatidylserine (PS) [159]. PS is a component present in the inner lobe of the plasma membrane of normal mammalian cells. The expression of PS in cancer cells is transferred to the outer lobe of the plasma membrane, resulting in a negative charge on the surface of cancer cells. Therefore, this chemical difference contributes to the electrostatic interaction between the AMPs and cancer cells, rapidly disrupting the cell membrane and causing the flow of cell contents to induce cell death. In addition, the increase in membrane surface area caused by microvilli on the membrane surface of cancer cells and the increase in membrane fluidity caused by reducing the level of lipoprotein in the membrane are more conducive to the binding of AMPs to cancer cells [160,161]. Of course, AMPs also disrupt membrane structure in a dose-dependent manner, with LfcinB (20–25)_4_, G3, melittin, HDH-LGBP-A1, and HDH-LGBP-A2 disrupting the cell membrane at high concentration levels [43,70,112,121].

### 3.5. Inhibition of Metastasis

Metastasis is the process by which cancer cells spread from the original tumor cells to nearby tissues or organs. The extracellular matrix (ECM) is an obstacle to tumor invasion and metastasis. In the process of cancer metastasis, a variety of proteases (matrix metalloproteinases (MMPs)) can degrade the ECM, which is conducive to the invasion and metastasis of cancer cells [162,163]. MMP downregulation has been reported to inhibit cancer metastasis [164,165]. At present, the research on inhibiting metastasis mainly focuses on the factors or pathways related to MMPs, among which MMP-2 and MMP-9 play a key role in tumor progression. Lactacin, melittin, cecropin, and rpNK-lysin can all inhibit cancer cell metastasis by downregulating key members of the MMP family [65,82,103,118]. Metastasis-associated protein 2 (MTA 2) is closely related to the progression of various cancers such as liver cancer, gastric cancer, and colorectal cancer [166,167]. Human β-defensin-3 (HBD3) inhibits cancer cell metastasis by downregulating MTA2 [168]. In addition, anti-metastasis mechanisms in cancer are closely related to cell signaling pathways. Human α-defensin 5 (DEFA5) attenuates the downstream signal transduction of the PI3K-AKT pathway by binding to subunits of the PI3K complex, resulting in delayed cell metastasis [166]. Human α-defensin 6 (HD6) inhibits colorectal cancer metastasis by regulating the EGF/EGFR signaling pathway [129]. MMP-2 and MMP-9 are the downstream target genes of the Wnt/β-catenin signaling pathway [169]. rpNK-lysin regulates the Wnt/β-catenin signaling pathway by downregulating Fascin1 and inducing β-catenin degradation and inhibits the expression of downstream target genes MMP-2 and MMP-9 [103].

## 4. The Effect of Modification and Optimization of Antimicrobial Peptides on Gastrointestinal Tumors

### 4.1. Peptide Length

Most natural AMPs have relatively long primary sequences, but it is usually the core amino acid fragments in AMPs that have biological activity. Long sequences are usually limited by high production costs and instability of enzymatic degradation. Cecropin B, an antimicrobial peptide composed of 35 amino acids, has shown a wide range of antitumor activities in previous studies, including inhibition of the proliferation of liver cancer cells, gastric cancer cells, and bladder cancer cells [170,171,172]. Chunli Wu et al. synthesized cecropin-p17, an analogue with the same net charge as cecropin B, based on an amphiphilic structural design, which consists of only 17 amino acids. Cecropin-P17 inhibited HepG-2 cells in a time- and dose-dependent manner, and showed low cytotoxicity on human normal liver L02 cells [99]. Elaheh Jamasbi et al. designed a branched chain dimer form of melittin and found that a short sequence melittin monomer was more toxic to gastric cancer cells than a long sequence dimer at low concentrations (1–5 μM) [69]. *Musca Domestica Cecropin* (MDC) is a linear molecule of 40 amino acids, with M1-8 derived from the N-terminal 1–8 amino acids of MDC. Previously, MDC has been shown to inhibit the growth of hepatocellular carcinoma cells. Jiali Zeng et al. found that M1-8 also showed excellent antiproliferation ability for hepatocellular carcinoma HepG2 cells and significantly inhibited the growth of tumors, indicating that short sequence peptide M1-8 did not seem to have any effect on hepatocellular carcinoma [82,95]. LL37 has an amphiphilic long helical structure spanning 2–31 residues, and FK-16 corresponding to 17–32 residues retains antibacterial and antitumor effects. Shun X. Ren et al. found that the cytotoxicity of short sequence FK16 on colon cancer cells was stronger than that of LL37, which showed better anticancer effects than the full-length peptide [113]. Notably, the effect of FK16 made cancer cells more susceptible to membrane disruption, suggesting that the FK16 fragment is a core functional region of LL37. Yahya Acil et al. found that KI-21-3, a shortened fragment of LL-37, exhibited the same anti-oral cancer mechanism as LL37, and the oncological effect of KI-21-3 was verified in vivo, indicating that the shortening of the peptide length did not affect the effect of KI-21-3, but could effectively solve the problem of high production costs faced with long sequences [45].

### 4.2. Peptide Charge

The cancer cell membrane is rich in glycoproteins and PS anions, so AMPs with more positive charges may act more effectively on cancer cell membranes. Bo-Hye Nam et al. found that HDH-LGBP-A2, which has one more net positive charge than HDH-LGBP-A1, increased the cytotoxicity of HeLa, A549, and HCT 116 cancer cells by 183.3%, 75%, and 45.5%, respectively, at low concentrations [121]. Several derived peptides of LfcinB have charges above +3. Víctor A. Solarte et al. found that LfcinB (20–25)_4_, a derived peptide with a net positive charge of +16, exhibited higher cytotoxicity in CAL27 and SCC15 cells, with inhibition rates of 93% and 96%, respectively [43]. Mengyun Ke et al. designed and synthesized a novel peptide Mel-PEP by replacing the valine at the eighth position and the proline at the fourteenth position of MEL with lysine; this modification increased the charge and helicity of the peptide. They found that Mel-PEP exhibited a stronger antiproliferation ability than MEL against liver cancer BEL-7402/5-FU cells [100]. In addition, it has been reported that highly charged amphiphilic peptides do not exhibit significant cytotoxicity [101]. Therefore, the charge does not positively correlate with the anticancer activity of the peptide, and there seems to be a threshold.

### 4.3. Peptide Secondary Structure

Secondary structure is an important determinant of protein function and activity. Most natural or synthetic AMPs have a certain secondary structure, such as α-helix, which may promote the formation of holes in the membrane of cancer cells, leading to the leakage of cell contents. It may also interfere with phospholipid fluidity and form transient pores in the membrane of cancer cells, prompting AMPs to enter the cell to play a role. The MDC mentioned in the peptide length showed three α-helical structures at residues 1–6, 9–21, and 27–39. M1-8 derived from the N-terminal 1-8 amino acids containing a helix structure in MDC appeared to have no effect on hepatocellular carcinoma HepG2 cells [95]. Therefore, it is speculated that the α-helix structure may be closely related to the anticancer effect. The RWL sequence is the C-terminal trimer of the chicken β-defensin AvBD-4, and H stands for the amino acid sequence GLRPKYS. N. Dong et al. designed H-(RWL) n (n = 1, 2, 3, 4, 5) peptides GL10, GL13, GL16, GL19, and GL22, among which GW10 and GW13 showed a disordered conformation and GW16, GW19, and GW22 showed a secondary structure with an α-helical conformation. Peptides with higher RWL content were richer in their α-helical structure. Compared with peptides GW10 and GW13, peptides with an α-helical conformation (GW16, GW19, and GW22) showed higher cytotoxicity on human hepatocellular carcinoma HepG2 cells [91]. This finding suggests that the abundant α-helical structure may be responsible for the gradual increase in cytotoxicity.

### 4.4. Combined and Coupled Peptides

At present, there are still challenges in the clinical use of AMPs. Coupling with polymers, coupling with small molecule peptides, or combination with anticancer drugs may overcome the shortcomings of AMPs and improve the therapeutic potential of AMPs. It has been reported that bacteriocin (enterocin-A +enterocin-B), 10 μg/mL HNP 1 and 50 μg/mL lactoferrin, colicin A and colicin E1, TMTP1 and DKK fusion peptide, MELITININ, and BMAP27 coupling peptide all showed significant anticancer activity against gastrointestinal tumors [48,63,66,75,125]. In addition, the combination of gramicidin A (GA) and the anticancer drug doxorubicin (Doxo) also significantly reduced cancer cell viability [139]. This class of combinatorial coupled peptides showed certain synergistic effects, and there may be a subtle relationship between them to inhibit the growth of cancer cells.

## 5. Advantages of Antimicrobial Peptides in Anti-Gastrointestinal Tumor Treatments

### 5.1. High Selectivity

AMPs, as short sequence peptides containing amino acids, are a better choice as tumor therapeutic agents compared with antibodies and small molecules because of their high selectivity. There are significant differences in cell membrane composition between healthy cells and cancer cells. Eukaryotic membranes contain large amounts of amphotericin phosphatidylcholine and cholesterol, while cancer cells contain higher amounts of anions such as O-glycosylation mucin, phosphatidylserine, and heparan sulfate [173,174,175]. Cancer cells have a high transmembrane potential compared with normal eukaryotic cells. Therefore, cationic AMPs mainly interact with normal eukaryotic cells via hydrophobic interaction but with cancer cell membranes via electrostatic interaction. For example, CopA3 mediates cell necrosis through specific interactions with cancer cell membrane phosphatidylserine and phosphatidylcholine [176]. Since the eukaryotic cell membrane is rich in cholesterol, this property can increase the cohesion of the lipid bilayer to prevent membrane disruption and can also change membrane fluidity to prevent membrane dissolution [177]. In addition, cancer cells also contain more abundant microvilli compared with healthy cells, which increases the membrane surface area and is more conducive to the interaction of AMPs with cancer cells [171,178].

### 5.2. Drug Resistance

Currently, conventional chemotherapy remains the preferred treatment for cancer, but its effectiveness often prevents intrinsic or acquired resistance. The mechanism of action of AMPs is less likely to lead to resistance than conventional chemotherapy [179,180,181]. Conventional chemotherapeutic drugs must enter the cell to work effectively, while AMPs can selectively attach to the membrane of cancer cells, thereby destroying the cell membrane and causing the cell content to flow out. Some AMPs exert their effect before entering the cell, and this unique mechanism is the possible reason for reduced resistance [9,182]. In addition, epithelial-to-mesenchymal transition (EMT) was detected to be closely related to drug resistance in hepatocellular carcinoma (HCC) [183]. This process is regulated by the transcription factors ZEB1, TWIST1, and SNAI1. Among them, the downregulation of TWIST1 leads to rapid cell death and increased sensitivity to drug treatment [184,185]. Pelin Balcik-Ercin et al. found that treatment of the hepatocarcinoma cell line HuH-7 with the bacteriocin lactacin resulted in significant inhibition of SNAI1 and TWIST1 expression, which are critical for drug resistance [104].

P-glycoprotein (P-gp) is one of the widely studied MDR proteins, also known as multidrug resistance protein 1 (MDR1), whose main function is to expel chemotherapy drugs from cancer cells [186,187]. Mengyun Ke et al. found that when the novel antimicrobial peptide MEL-pep was used in human 5-FU-resistant HCC cells (BEL-7402/5-FU), it could inhibit the expression of P-gp by inhibiting the PI3K/Akt pathway to improve the sensitivity to 5-FU, which has great potential in the treatment of drug-resistant cancer [100].

## 6. Conclusions and Prospects

Cancer is the main cause of death in the world population. The burden of cancer is increasing, and the prevention and treatment of cancer are facing a serious challenge. Traditional cancer therapies have certain drawbacks, such as drug side effects, low specificity, and drug resistance of cancer cells. Therefore, the development and use of AMPs have become a new means for the treatment of cancer. In this review, we discuss the anti-gastrointestinal tumor mechanisms of AMPs, their limitations as anticancer drugs, their specificity to tumor cells, and their sensitivity. Due to their specificity and sensitivity to tumor cells, some AMPs have been shown to have potential therapeutic effects in different types of gastrointestinal tumors.

Although the anticancer potential of many AMPs has been proved, few AMPs have been used in clinical treatment, and the clinical application and development of AMPs still face great challenges. First, the yield of natural AMPs is low and the extraction procedure is complex, while the high price of synthetic AMPs is not suitable for their commercial development. AMP can be abundantly expressed in heterologous expression systems of microbial cells, and heterologous expression technology holds promise for improving AMP production. Second, AMP is exceptionally sensitive to degradation by proteases and is easily cleaved by proteases in vivo and rapidly excreted from the kidney, resulting in its short half-life. Chemical modification methods such as sequence manipulation, net charge, and secondary structure are beneficial for solving the limitations of AMPs, such as poor stability, low bioavailability, and proteolytic enzyme degradation. The combination of AMPs and AMPs, AMPs and currently used chemotherapy drugs, and AMPs and nanocarriers can also help to improve the pharmacokinetics, half-life, bioavailability, and targeting specificity of AMPs, and reduce the side effects in patients.

In conclusion, AMPs are potential drugs for cancer treatment, but more comprehensive and in-depth research is needed to make them better and more efficient for clinical application.

## Figures and Tables

**Figure 1 ijms-24-16718-f001:**
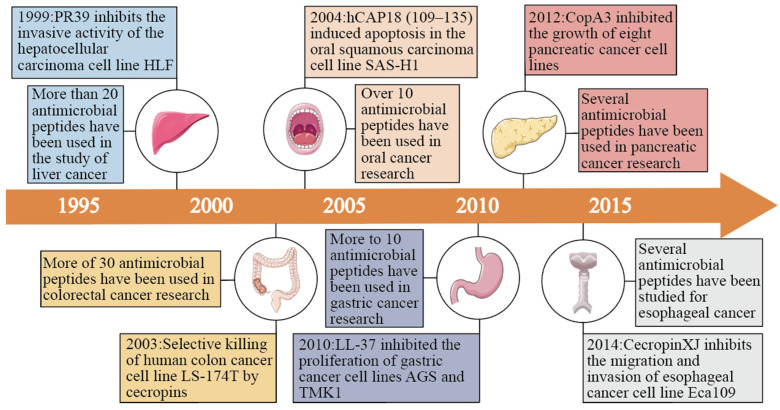
Early studies of antimicrobial peptides in various gastrointestinal cancers (oral, esophageal, gastric, liver, pancreatic, and colorectal) and the number of antimicrobial peptides studied to date in various gastrointestinal cancers.

**Figure 2 ijms-24-16718-f002:**
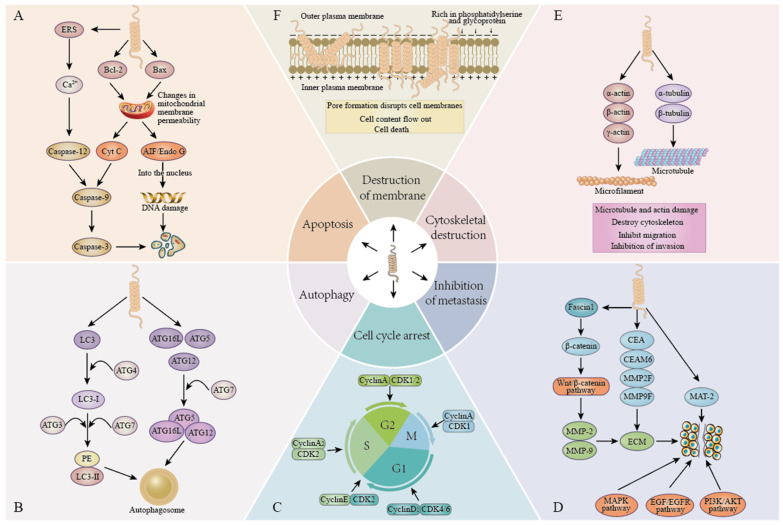
Anticancer mechanisms of antimicrobial peptides. (**A**). Apoptosis: the induction of apoptosis in cancer cells by either caspase-dependent or caspase-independent pathways. (**B**). Autophagy: inducing autophagy in cancer cells by regulating the expression of autophagy markers and autophagy-related proteins. (**C**). Cell cycle arrest: results in cell cycle arrest by regulating the expression of cyclin and cyclin-dependent kinase (CDK). (**D**). Inhibition of metastasis: inhibition of cancer cell metastasis by regulating signaling pathways and inhibiting the expression of matrix metalloproteinases. (**E**). Destruction of the cytoskeleton: by regulating the expression of actin and tubulin, microtubules and microfilaments are damaged and the cytoskeleton is destroyed. (**F**). Membrane destruction: the pore formation mechanism destroys the cell membrane and causes the cell content to flow out, leading to cell death.

**Table 1 ijms-24-16718-t001:** Examples of antimicrobial peptides used in the treatment of oral cancer and their properties.

AMP	Sequence	Source	Cell Line	Cytotoxicity (IC50)	Mechanism of Action	Reference
Pardaxin	H-GFFALIPKIIS SPLFKTLLSAVGSALSSSGGQE-OH	*Pardachirus marmoratus*	SCC-4	Pardaxin (5, 10, 15, 20, and 25 μg/mL) inhibits the growth rate at 24 and 48 h after treatment	Caspase-3 activation-induced apoptosis and G2/M phase-induced cell arrest	[42]
LfcinB(20–25)4	(RRWQWR)4-K2-(Ahx)2-C2	The tetrameric peptide of bovine lactoferrin	CAL27, SCC15	IC50 for CAL27 = 9.016 ± 1.38 Μm, IC50 for SCC15 = 9.048 ± 1.07 μM	Apoptosis occurs at low concentrations and cell membrane necrosis occurs at high concentrations	[43]
CDEL/CAMP	/	/	HSC-3	After 24 h, 48 h, and 72 h of treatment, cell proliferation is inhibited	Activation of the P53-Bcl-2/BAX signaling pathway induces caspase-3-mediated apoptosis	[44]
KI-21-3	KIGKFFKRIVRIKKFIRKFV-NH 2	LL-37	SCC-4	In vivo, the tumor weight is reduced by 30% compared with the control group, and the volume changes a little	Apoptosis induced by antiproliferation and caspase-3	[45]
hCAP18 (109–135)	FRKSKEKIGKEFKRIVQRIKDFLRNLV	C-terminal domain of hCAP18	SAS-H1	The cytotoxicity is 14 ± 3.2% at 48 h and 80 ± 5.3% at 96 h, respectively	Induces apoptotic cell death and oligosomal DNA fragmentation through caspase-independent pathways	[46]
HNP-1	/	Human polymorphonuclear leukocytes (neutrophils)	UT-SCC-43A, UT-SCC-43B	After 48 h of treatment, HNP1 (1–10 μg/mL) has no obvious cytotoxicity	/	[47]
HNP-1 + lactoferrin	/	/	Two oral squamous cell carcinoma (OSCC) lines	The cytotoxicity of lactoferrin (12.5–100 μg/mL) increases at 72 h after treatment. HNP-1 (100 μg/mL) has significant cytotoxicity at 24, 48, and 72 h after treatment	Lactoferrin (50 μg/mL) and HNP-1 (10 μg/mL) show selective oncolytic effects	[48]
HBD-1	/	Epithelial tissue	BHY-OSCC, HSC-3, UM1, SCC-9, SCC25	After 24 h of treatment, HBD-1 (50 nM) reduces the proliferation of BHY-OSCC cells by 25%. HBD-1 (50 mg/mL) does not significantly inhibit the proliferation of HSC-3/UM1/SCC-9/SCC25 cells	HBD-1 may be a tumor suppressor gene in oral squamous cell carcinoma. Exogenous expression of HBD-1 significantly inhibits migration and invasion	[49,50]
NRC-03	GRRKRKWLRRIGKGVKIIGGAALDHL-NH2	Skin mucous secretions of winter flounder	CAL-27, SCC-9	The cytotoxicity of NRC-03 (15–75 μg/mL) is significantly increased at 4 h after treatment	The cypD-mPTP axis mediates mitochondrial oxidative stress-induced apoptosis	[51]

**Table 2 ijms-24-16718-t002:** Examples of antimicrobial peptides used in the treatment of esophageal cancer and their properties.

AMP	Sequence	Source	Cell Line	Cytotoxicity (IC50)	Mechanism of Action	Reference
LvHemB1	DVNFLLHKIYGNIRY	N-terminal domain of L. vannamei hemocyanin	EC190	After 24 h of treatment, the viability of LvHemB1 (50 µg/mL) cells decreased by 49.1%	Antiproliferative effects and targeting of the voltage-dependent anion channel 1 (VDAC1) lead to mitochondrial dysfunction in cancer cells, as well as the induction of apoptosis by increasing ROS levels, and the expression of proapoptotic proteins	[58]
BmCecA and BmCecD	BmCecA: RWKLFKKIEKVGRNVRD GLIKAGPAIAVIGQAKSLGK BmCecD: GNFFKDLEKMGQRVRDAVISAAPAVDTLAKAKALGQ	*Bombyx mori*	Eca109, TE13	After 12 h of treatment, the inhibition rates of BmCecA (100 µg/mL) EC190 cells and TE13 cells are 36.68 ± 2.31% and 17.71 ± 2.81%, respectively. BmCecD (100 µg/mL) decreases the viability of EC190 cells by 30.72 ± 1.62%, and the inhibition rate of TE13 cells is 21.92 ± 3.7%	BmCecA induces apoptosis of Eca109 cells by activating the mitochondria-mediated caspase pathway, upregulating Bcl-2-associated X protein, and downregulating Bcl-2	[54]
Cecropin A	RWKLFKKIEKVGRNVRDGL IKAGPAIAVIGQAKSLGK	*Bombyx mori*			By binding to the mitochondrial membrane and entering the cytoplasm, damage to the mitochondrial membrane triggers apoptosis	[55]
CecropinXJ	MNFAKILSFVFALVLALSMTSAAPEPRWKIFKKIEKMGRNIRDGIVKAGPAIEVLGSAKAIGK	*Bombyx mori*	Eca109		Cytotoxicity is triggered by inducing cytoskeletal disruption and regulating the expression of cytoskeletal proteins	[57]
Cecropin D	GNFFKDLEKMGQRVRDAV ISAAPAVDTLAKAKALGQ	*Bombyx mori*			By penetrating deeply into the mitochondrial membrane bilayer containing cardiolipin, it leads to a significant destabilization of lipid packaging, which may account for its proapoptotic activity	[56]

**Table 3 ijms-24-16718-t003:** Examples of antimicrobial peptides used in the treatment of gastric cancer and their properties.

AMP	Sequence	Source	Cell Line	Cytotoxicity (IC50)	Mechanism of Action	Reference
Enterocin A–Colicin E1	/	*E. coli*	AGS	Treatment for 24 h IC50 = 60.41 μM Treatment for 48 h IC50 = 48.71 μM	The Bax/bcl-2 ratio at the mRNA level is increased to induce apoptosis	[63]
Bovine lactoferricin B	FKCRRWQWRMK KLGAPSITCVRRAF	Fragments of bLF pepsin hydrolysis	AGS, 3T3	IC50 for AGS = 64 μM IC50 for 3T3 ≥ 500 μM	Increasing mRNA levels inhibits the final stage of autophagy and enhances the bax/bcl-2 ratio of caspase-dependent apoptosis to induce apoptosis	[64]
Melittin	/	Bee venom	AGS	Melittin (0.2–0.5 μM) reduces the number of viable cells by 24–79% at 24 h after treatment	Downregulating the expression of vimentin, N-cadherin, and MMP-2 and upregulating the expression of E-cadherin, MMP-9, and MMP-13 inhibits metastasis through Wnt/β-catenin, BMP/Smad, and EMT signaling pathways	[65]
Enterocin-B and enterocin-A + B	/	*E. faecium por1*	AGS	After 24 h of treatment, the inhibition rate of enterocin-B (25 μg/mL) is 22.84 ± 2.68%. After 24 h of treatment, the inhibition rate of enterocin-A+B (25 μg/mL) is 51.76 ± 1.12%	/	[66]
GW-H1	GYNYAKKLAN LAKKFANALW		AGS, 3T3	IC50 for AGS = 17 μM IC50 for 3T3 = 243 μM	Apoptosis and autophagy are induced in the AGS cell line at the early stage, and caspase-dependent apoptosis is further enhanced by inhibition of autophagy at the late stage	[67]
CopA3	/	An analog derived from coprisin	SNU-484, 601, 638, 668	IC50 for SNU = 18–29 μM IC50 for human keratinocytes ≥ 500 μM	Induction of caspase-dependent pathway apoptosis and necrosis in gastric cancer in vitro and in vivo increases selective toxicity through specific interactions with cancer cell membrane phosphatidylserine and phosphatidylcholine	[68]
Dimerized melittin	GIGAVLKVLTTGLPALISWIKRKRQQ-Dab-NH2	Bee venom	NUGC-3, MKN-7, MKN-74	Dimerized melittin (1–5 μM) is highly cytotoxic in gastric cancer cells after 24 h of treatment	Both melittin monomers and dimers penetrated the cytoplasm	[69]
Melittin	/	Bee venom	AGS	The survival rate of melittin (5–20 μg/mL) is significantly decreased at 4 h after treatment	A high dose of melittin has a membrane effect on gastric cancer cells over a time course of 15 min, with cellular changes occurring within seconds in the form of cell swelling, membrane blebbing, and fragmentation	[70]
DEFA5	/		SGC7901, HEK293T, BGC823	After 24 h of treatment, overexpression of human DEFA5 effectively reduces cell proliferation and colony formation ability	It inhibited cell proliferation by directly binding to BMI1, reducing its binding to CDKN2a, and upregulating the expression of two cyclin-dependent kinase inhibitors, p16 and p19	[71]
LL-37	/	Human	AGS, TMK1	The inhibition rate of LL-37 (25 mg/mL) at 24 h of treatment is 60% for TMK1 and 10% for AGS	Activation of the BMP signaling pathway inhibits cell proliferation through a proteasome-dependent mechanism	[72]
Melittin	/	Bee venom	SGC-7901	/	It induces the release of ROS and the opening of the mitochondrial permeability transition pore, releases Cyt C, Smac/Diablo, AIF, and EndoG proteins, activates caspase-3, leads to the formation of apoptotic bodies, and ultimately produces apoptosis	[61]
CecropinXJ	WKIFKKIEKMGRNIRDGIVKAGPAIEVLGSAKAIGK	*Bombyx mori*	BGC823	The proliferation of BGC 823 cells is inhibited in a dose- and time-dependent manner	It inhibits cell growth in vitro and in vivo by promoting ROS production, reducing mitochondrial membrane potential, inducing apoptosis in the caspase pathway, and preventing tumor angiogenesis	[73]
Melittin	/	*Iranian honeybee venom*	AGS	The proliferation of AGS cells is inhibited in a dose- and time-dependent manner	Melittin has an anticancer effect on gastric cancer AGS cells and stimulates necrotic cell death in these cells	[74]
TMTP1-DKK	KLAKLAKK LAKLAK		MKN-45	/	Apoptosis is triggered in a series of highly metastatic cancer cells via the mitochondrial pathway and the death receptor pathway	[75]

**Table 4 ijms-24-16718-t004:** Examples of antimicrobial peptides used in the treatment of liver cancer and their properties.

AMP	Sequence	Source	Cell Line	Cytotoxicity (IC50)	Mechanism of Action	Reference
DEFB1	/	Epithelial tissues	HepG, Huh7, HCCLM3	Overexpression of DEFB1 decreased cell proliferation in a time-dependent manner	Activation of the JNK pathway induced by ER stress exerts an inhibitory effect on cell proliferation during tumor growth	[80]
MzDef	MSSSNCANVCQTENFPGGECKAEGATRKCFCKNC	*Maize*	HePG2	IC50 = 14.85~29.85 μg/mL		[81]
Cecropin	MNFNKLFVFVALVLAVCIGQSEAGWLKKIGKKIERVGQHTRdATIQTIGVAQQAANVAATLKG	*Musca domestica*	BEL-7402	/	By disrupting the microvilli of tumor cells and altering the expression of MMP2, TIMP2, and E-cadherin, cell adhesion and migration are inhibited	[82]
Cecropin	MNFNKLFVFVALVLAVCIGQSEAGW- LKKIGKKIERVGQHTRDATIQTIGVAQQAANVAATLKG	*Musca domestica*	BEL-7402	Cecropin (12.5–100 mM) inhibits the proliferation of BEL-7402 cells in a dose- and time-dependent manner	It may induce cell apoptosis by upregulating the expression of Fas, Fas-L, caspase-8, and caspase-3 and triggering the extrinsic apoptotic pathway	[83]
SK84		*Drosophila*	HePG2	IC50 = 92 μg/mL		[84]
CecropinXJ	/	*Bombyx mori*	Huh7	The inhibition rate of cecropinXJ (50 µmol/L) is 36.6 ± 0.1%, which inhibits cell proliferation in a dose- and time-dependent manner	Inhibition of cell proliferation and induction of apoptosis in vitro via mitochondrial apoptotic pathways including loss of Δψm, the release of mitochondrial cytochrome c, and activation of caspase-3 and PARP	[85]
rCec-B	/	*Drosophila melanogaster*	HePG2	IC50 for HePG2 =25 μg/mL		[86]
GW-H1	/	Synthesis	J5, Hep3B, Huh7	IC50 for J5 = 20.3 μg/mL IC50 for Hep3B = 67.2 μg/mL IC50 for Huh7 = 87.2 μg/mL IC50 for 3T3 = 234.3 μg/mL	Caspase-dependent apoptosis is induced	[87]
Smp43	/	*Egyptian scorpion Scorpio maurus palmatus*	HepG2, Huh7	IC50 for HepG2 = 4.69 μg/mL IC50 for Huh7 = 5.14 μg/mL	Internalization into cells through endocytosis and pore formation leads to mitochondrial dysfunction and cell membrane disruption, inducing apoptosis, autophagy, necrosis, and cell cycle arrest	[88]
B11	RIRDAIAHGYIVDKV	Copper-containing domain of L. vannamei hemocyanin	HePG2	After 24 h of treatment, the viability of B11 (50 µg/mL) cells decreased by 23.0%	It has an antiproliferative effect on cancer cells, can cause mitochondrial dysfunction, and induces apoptosis	[89]
Brevinin-1BYa	FLPILASLAAKFGPKLFCLVTKKC	*Frog Rana boylii*	HePG2	LC50 =6 μg/mL		[90]
GW13	GLRPKYS(RWL)2-NH2	Chicken epithelial tissue	HePG2	The survival rate of MRC-5 cells treated with GW13 (128 μM) for 24 h is 81%, while that of HepG2 cells is 3%	Cell death is induced by pore formation and selective membrane disruption, as well as apoptosis	[91]
Bombinin-BO1 and Bombinin H-BO1	GIGSAILSAGKSIIKGLAKGLAEHF-NH2 and IIGPVLGLVGKALGGLL-NH2	*Bombina orientalis*	HepG2, SKHEP-1, Huh7	Bombinin-BO1: IC50 for SKHEP-1 = 0.76 μg/mL IC50 for Hep G2 = 3.75 μg/mL IC50 for Huh7 = 3.91 μg/mL Bombinin H-BO1: IC50 for SKHEP-1 = 3.61 μg/mL IC50 for Hep G2 = 8.08 μg/mL IC50 for Huh7 = 8.42 μg/mL	/	[92]
LL-37	LLGDFFRKSKEKIGKEFKRIVQRIKDFL RNLVPRTES	Human	HepG2, Huh7	LL-37 (10–20 μM) significantly reduced the viability of Huh7 and HepG2 cells after 48 h of treatment	By inhibiting the CyclinD1-CDK4-p21 checkpoint signaling pathway, it delays the G1-S transition in cells and participates in apoptosis and proinflammatory cytokine production	[93]
Trichokonin VI		*Trichoderma pseudokoningii SMF2*	HCC	/	Promotes Ca^2+^ influx, activates calpain, then cleaves Atg5 and Bax, combines with Bcl-xL to destroy mitochondria, accelerates the destruction of mitochondrial integrity and cytochrome c release, further activates caspease3, and eventually leads to apoptosis. Ca^2+^ influx activates BaK and promotes ROS accumulation, and ROS-susceptible cells undergo autophagy through the mitophagy pathway	[94]
M1-8	GWLKKIGK	*Musca domestica cecropin*	HepG2	After 24 h of treatment, M1-8 (25 μg/mL) inhibited the proliferation of HepG2 cells	Upon exposure to M1-8, human hepatocellular carcinoma HepG2 cells rapidly colocalize with lysosomes, destroying lysosomal integrity and blocking autophagy–lysosome fusion, leading to leakage of lysosomal protease cathepsin D, activation of caspases, and changes in mitochondrial membrane potential, and promoting apoptosis.	[95]
Nisin		*Streptococcus* spp and *Lactococcus* spp	HePG2	IC50 = 40 μg/mL	It plays a role in apoptosis by increasing the cellular mitochondrial pathway	[96]
Smp24		Scorpio Maurus palmatus	HepG2	IC50 for HepG2 = 5.52 μg/mL IC50 for LO2 = 16.68 μg/mL	Entry into cells through pore formation and endocytosis leads to mitochondrial dysfunction and membrane defects, which lead to cell necrosis, cycle arrest, apoptosis, and autophagy	[97]
PR-39		Pig small intestines	Huh1, Huh2, HL E, HLF		Induction of Syndecan-1, inhibition of invasion and motility activity, and changes in actin structure	[98]
Cecropin-P17	FKKKVGRNIRNGIIK	Cecropin B	HepG2	The survival rate of cecropin-P17 (40 μg/mL) cells is 45.3% at 48 h after treatment	By increasing the concentration of ROS in cells, it activates caspase-3 and caspase-9, increases the expression of Bcl-2 protein, decreases the expression of Bax protein, promotes cell apoptosis, and inhibits cell proliferation in vitro and in vivo	[99]
Melittin (MEL) and MEL-pep	MEL-pep: GIGAVLKKLTTGLKALISWIKRKRQQMEL: GIGAVLKVLTTGLPALISWIKRKRQQ	*Bee venom*	BEL-7402/5-FU	MEL-pep: IC50 for BEL-7402/5-FU = 4.44 μg/mL MEL: IC50 for BEL-7402/5-FU = 11.09 μg/mL	MEL-pep could significantly inhibit the proliferation of BEL-7402/5-FU cells by selectively binding to and destroying the cell membrane. MEL-pep could also reverse the drug resistance of BEL-7402/5-FU cells and restore the sensitivity to 5-FU	[100]
WRL3	WLRAFRRLVRRLARGLRR-NH2	*Leuconostoc gelidum* UAL 187	HepG2	IC50 for HepG2 = 32 μg/mL IC50 for LO2 = 16.68 μg/mL	The killing of microorganisms and tumor cells by disrupting the cell membrane leads to cytoplasmic efflux	[101]
HBD-3	/	Epithelial cells	Huh7.5	/	Activated PBMC secretes IFN-γ and kills K562 and HUH liver cancer target cells in an NK-dependent manner, and both TLR1/2 and CCR2 are involved	[102]
rpNK-lysin	/	Porcine intestinal tissue	HepG2, SMMC-7721, MHCC 97-H	MNTC for LO2 = 90.8 μg/mL MNTC for SMMC-7721 = 54.16 μg/mL MNTC for MHCC 97-H = 51.76 μg/mL MNTC for HepG = 47.38 μg/mL MNTC = maximum nontoxic concentration	Fascin1 inhibits the invasion and metastasis of HCC cells by downregulating Fascin1. Fascin1 further regulates the Wnt/β-catenin signaling pathway, induces β-catenin degradation, and inhibits the expression of MMP-2 and MMP9	[103]
Nisin		Lactococcus and streptococcus species	SNU182, Huh7	At 24 and 48 h after treatment, Nisin (48–160 μg/mL) begins to inhibit proliferation	It has a potent antitumor effect on HCC by reducing cell proliferation and activating apoptosis in HCC disease model cell lines	[104]
CopA3	LLCIALRKK-NH2	*Copris tripartitus*	Hep3B, Hep2G, SK-Hep1, SNU-182, SNU-354	IC50 = 67.8 µM	/	[105]

**Table 5 ijms-24-16718-t005:** Examples of antimicrobial peptides used in the treatment of colorectal cancer and their properties.

AMP	Sequence	Source	Cell Line	Cytotoxicity (IC50)	Mechanism of Action	Reference
rSs-arasin	MERRTLLIVLLVCSFLLLAVTAEA	*Scylla serrata*	HT-29	IC50 = 2.90 μM	/	[109]
Plantaricin P1053	/	*L.plantarum* PBS067 strain	E705	Phytomycin P1053 (1 μg/mL) inhibited cell proliferation by about 30% at 48 h after treatment	/	[110]
GA-W3 and GA-W4 and GA-K3 and GA-K4	FLGWLFKWAWK-NH2 and FLWWLFKWAWK-NH2 and FLGWLFKWAKK-NH2 and FLKWLFWAKK-NH2	Brevinin-1EMa	HCT-116	GA-W3: IC50 = 24.63 μM GA-W4: IC50 = 14.80 μM GA-K3: IC50 = 27.00 μM GA-K4: IC50 = 14.80 μM	/	[111]
G3	G(IIKK)_3_I-NH2		HCT-116	The cytotoxicity of G3 (100 μM) is 80%, 85%, and 95% at 24 h, 48 h, and 72 h after treatment, respectively	High concentrations of peptides disrupt tumor cell membranes	[112]
FK-16	FKRIVQRIKDFLRNLV	LL-37	LoVo, HCT-116	The cytotoxicity of FK-16 (40 μM) LoVo cells is about 50%, and that of HCT116 cells is about 60%	Activation of p53 induces caspase-independent apoptosis and autophagic cell death in colon cancer cells by upregulating Bax and downregulating bcl-2	[113]
HPA3P	AKKVFKRLPKLFSKIWNWK-NH2	Analogs of *Helicobacter pylori ribosomal* protein L1	LoVo, HT-29, SW-480, HCT-116 p53+/HCT-116 p53-/	After 6 h of treatment, HPA3P (60 μM) has about 80% cytotoxicity in all types of cells	Ripk3-dependent necroptosis is induced	[114,115]
Enterocin-A	MKHLKILSIKETQLIYGGTTHSGKYYGNGVYCTKNKCTVDWAKATTCIAGMSIGGFLGGAIPGKC	*Escherichia faecalis* Por1	HT-29	The cytotoxicity of enterocin-A (120 μg/mL) at 24 h and 48 h is 56.16 ± 0.41% and 83.74 ± 0.47%, respectively	Sub-G and G1 phase cell cycle arrest as well as induction of apoptosis and cell death	[116]
GM3	/	Goat milk	HT-29	The cytotoxicity of GM3 (2240 AU/mL) is 45.6 ± 0.6% at 24 h after treatment	/	[117]
Nisin	/	*Lactococcus lactis* subsp	LS-180, SW-48, HT-29, CaCO-2	After 24 h of treatment, Nisin (80–400 IU/mL) has about 50% cytotoxicity on LS-180 cells. Nisin (350–800 IU/mL) is approximately 50% cytotoxic to SW-48, HT-29, and Caco-2 cells	Reducing the expression of CEA, CEAM6, MMP2F, and MMP9F genes inhibits the metastasis of colon cancer cells	[118]
KL15	KRKLYKWFAHLIKGL	Bacteriocin m2163 and m2386 sequences	SW-480, Caco-2	IC 50 = 26.3 μM	Disruption of the cell membrane enhances membrane permeability to induce cell necrosis pathways	[119]
KT2	NGVQPKYKWWKWWKKWW-NH2	Crocodile white blood cells	HCT-116	IC50 = 50 μM	It promotes cell membrane defects and caspase-dependent pathways to mediate apoptosis and inhibit autophagy	[120]
HDH-LGBP-A1 and HDH-LGBP-A2	WLWKAIWKLLT-NH2/WLWKAIWKLLK-NH2	*Haliotis discus hannai*	HCT-116	The cytotoxicity of HDH-LGBP-A1 (25 μg/mL) and HDH-LGBP-A2 (25 μg/mL) is 93.96% and 93.6%, respectively	Cell detachment, swelling, and damage are induced, and disruption of the cell membrane leads to cell death	[121]
HD-5	ATCYCRTGRCATRESLSGVCEISGRLYRLCCR	Human intestinal tract	/	/	Disruption of cell membrane integrity and induction of apoptosis	[122]
LL-37	/	Epithelial tissue	p53 wild-type (HCT-116, LoVo) mutant (SW-1116, SW-620, SW-480)	LL37 (60 μmol/L) showed certain cytotoxicity on all cells	Nuclear translocation of AIF and EndoG through upregulation of Bax and Bak and downregulation of Bcl-2 induces caspase-independent apoptosis in colon cancer	[123]
RT2	NGVQPKYRWWRWWRWWW-NH2	Crocodile white blood cells	Caco-2	RT2 (120 μM) has 91.45% cytotoxicity	Inhibition of colon cancer cell proliferation by enhancing STARD13, TLE3, and OGDHL expression	[124]
MELITININ + BMAP27	/	/	HT-29, SW-742, HCT-116, WiDr	The IC50 is 30, 20, and 10 μg/mL at 24, 48, and 72 h of treatment, respectively	Apoptosis and autophagy mechanisms induce cancer cell death	[125]
CopA3	/	Coprisin	HCT-116, KM12C, ROK	After 96 h of treatment, CopA3 (5 μM) significantly reduced the number of cancer cells	Inhibits the growth and proliferation of colorectal cancer cells by inducing cell cycle arrest through an ROS-mediated pathway	[126]
Br-J-I	PF^a^KLSLHL-NH2	Royal jelly of honeybees	HCT-116, Lovo, HT-29, MC-38	Br-J-I (80 μM) treatment does not cause significant cytotoxicity at 72 h of treatment	It is not cytotoxic to cancer cells, but it can indirectly and effectively inhibit *Fn*-induced colorectal cancer and inflammatory protumor effects by killing *Fn*	[127]
m2163 and m2386	KRKCPKTPFDNTPGAWFAHLILGC and DSIRDVSPTFNKIRRWFDGLFK	LAB *L. casei* ATCC 334	SW-480, Caco-2	M2163: IC50 = 40 μg/mL M2386: IC50 = 40 μg/mL	The ratio of proapoptotic Bax/antiapoptotic protein Bcl-2 was altered to induce exogenous and endogenous apoptosis	[128]
HD6	/	Paneth cells in the small intestine	CaCO-2, HT-29, HCT-116, DLD-1	HD6 inhibited CRC proliferation	Inhibition of CRC proliferation and metastasis by eliminating EGF/EGFR signaling pathway	[129]
MzDef	MSSSNCANVCQTENFPGGECKAEGATRKCFCKNC	*Zea Mays* L.	HCT-116	Treatment for 24 h, IC50 = 14.85–29.85 μg/mL	/	[81]
CSA-13	/	LL-37	HCT-116	The cytotoxicity of CSA-13 (10 mg/mL) wild-type HCT 116 cells was 40–70%, and that of p53 null mutant was 60–70%	Induced cell cycle arrest and antiproliferation in wild-type and p53 deletion mutant HCT116 colon cancer cells	[130]
FF/CAP18	FRKSKEKIGKFFKRIVQRIFDFLRNLV	LL-37	HCT-116	FF/CAP18 (10 μg/mL) could inhibit the growth of HCT-116 cells	Induction of partial mitochondrial membrane depolarization, an early stage of apoptosis, has an antiproliferative effect on human colon cancer cell line HCT116	[131]
BO18	RGNWKVKYLRIIKNRGSF	*Oplegnathus fasciatus*	HT-29	After 24 h of treatment, the viability of HT-29 decreased, and with the increase in BO18 concentration, the inhibition ratio of HT-29 increased significantly	/	[132]
Nisin	/	*Lactococcus lactis*	SW-48	The cytotoxicity of Nisin (4000, 3000, 2500, 2000, and 1000 μg/mL) is 15%, 42.94%, 41.77%, 41.55%, and 79.22%, respectively, at 24 h after treatment	Cell intrinsic pathway apoptosis is induced by increasing the bax/bcl-2 ratio at both mRNA and protein levels	[133]
Lactoferrin	/	Cow’s milk	CaCo-2	Lactoferrin (0.02 μM, 0.2 μM or 2.0 μM) decreased cell proliferation at 24, 48, and 72 h of treatment	It prolongs the S phase of the cell cycle, resulting in a decrease in the cell proliferation rate	[134]
Bmattacin2	/	*Bombyx mori*	HCT-116	At 24 h of treatment, Bmattacin2 (12 μM) selectively killed HCT-116	/	[135]
BLf and LfcinB	LfcinB: FKCRRWQWRMKKLGAPSITCVRRAF	Pepsin proteolytic production	HT-29	Cytotoxicity is demonstrated at 50, 100, 200, 400, or 800 μg/mL at 4, 12, 24, or 48 h of treatment, respectively	It exerts antitumor activity on human colorectal cancer cells by activating various signaling pathways (p53, apoptosis, and angiogenin signaling)	[136]
MccE492	/	A bacteriocin produced by *Klebsiella pneumonia*	HT-29	The viability of MccE492 (30 μg/mL or 60 μg/mL) cells decreased to 66.4% or 50% at 24 h after treatment, respectively	/	[137]
rhCGA-N46	/	Human chromogranin A	HCT-116	IC 50= 1.997 μg/mL	Apoptosis of HCT-116 cells is induced via upregulation of BID and CAS-8 apoptotic genes via downregulation of oncogene BCL2 and upregulation of qPCR	[138]
Gramicidin A (GA)	VGALAVVV WLWLWLW	*Aneurinibacillus migulanus*	HT-29	IC50 = 9.78 μM	/	[139]

## Data Availability

Data is contained within the article. The data presented in this study are available in insert article. Data sharing is not applicable to this article.

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
