# Peer review of "Application Value of Antimicrobial Peptides in Gastrointestinal Tumors"

_ijms, 2023, doi:10.3390/ijms242316718_

Round 1
Reviewer 1 Report
Comments and Suggestions for Authors
The review paper from Qi Liu et al. describes the antitumorigenic properties exhibited by antimicrobial peptides (AMPs), especially as inhibitors of gastrointestinal tumor proliferation. AMPs are considered in the review in the broader sense, including eukaryotic AMPs, bacteriocins from Gram-positive and -negative bacteria, and marginally a few non ribosomal (trichokonin and gramicidin) or venom-derived (bee venom, scorpion venom) AMPs. Such a scope therefore includes ribosomally synthesized and posttranslationally modified peptides (RiPPs), examplified by some of the described AMPs (MccE492 from Klebsiella pneumoniae) or peptides arising from the NRPS pathway (trikoningin from Trichoderma koningii).
The review considers the different types of gastrointestinal tumors (oral and oropharyngeal, aesophageal, gastric, pancreatic, colorectal, liver, bladder tumors), which are briefly described. The origin and characteristics of the different AMPs acting on the different cancer types, the mechanisms of antitumor activity of the selected AMPs, the potential of engineering AMPs to optimize them for antitumor activity (peptide length, charge and secondary structure), and the advantages of AMPs compared to other compounds (antibodies, chemical compounds), including the important aspect of drug resistance, are then described successively. The review is supported by 182 literature references and illustrated and completed by synthetic Tables and Figures.
This is and interesting and original review in the field of AMPs, which is well presented and written in a good English. However, it is puzzled with some mistakes or incorrect assertions that have to be corrected and a few complementary discussion points should be added.
The points provided below have to be taken into account.
Specific comments
1- Introduction, lines 28-32. The definition provided for AMPs essentially applies to eukaryotic AMPs, which are most often unmodified cationic ribosomally synthesized peptides, with amphipathic properties, have a broad spectrum of activity and act on bacteria thanks to an interaction with anionic bacterial membranes and membrane perturbing properties, although some of them have more specific targets.
However (and this is fully pertinent), the review also takes into account and describes the properties of AMPs from bacteria (Gram-positive and Gram-negative bacteriocins and microcins) including those of ribosomal and non ribosomal origin. Therefore the definition provided in the paper by the authors is too restrictive. It has to include both unmodified ribosomally synthesized peptides and ribosomally synthesized and posttranslationally modified peptides (RiPPs), and marginally those arising from the NRPS (non ribosomal peptide synthetase) pathway, as very few peptides of this origin are concerned (trichokonin and gramicidin). The beginning of the introduction has thus to be rewritten (and appropriate references included) to describe the different types of AMPs depending on the producer organisms and biosynthetic pathways. In the line, change “Most AMPs” to “Many AMPs” line 296.
2- Introduction, lines 43-44. The authors do not consider immunotherapy in their paper, which is however currently developed as an expanding treatment for many kinds of gastrointestinal cancers, including liver and gastric cancers, either alone or in combination with chemotherapy. As an advantage, immunotherapy has fewer side effects compared to chemical treatments and radiotherapy that are cited in the introduction and discussed further in the paper (lines 431, 464-471). Omitting the potential of immunotherapy treatments is a serious lack in the review and description and comments on immunotherapy have to be included to make it fully comprehensive. Moreover, this will not depreciate the interest of AMPs as anticancer drugs in the future, as both approaches are needed.
3- Each table should use a homogeneous style for the different columns or lines (verb or name to describe the activity or function, tense (preterit or present,…) etc.
4- Throughout the Tables/Figures and the main text, please suppress “artificial synthesis”, which is an incorrect term; change it to “chemical synthesis”, “peptide synthesis” or “synthetic peptide”, depending on the context (sentence, table…).In addition peptide synthesis is not the origin of the peptide but the method to obtain it… It could be either designed completely de novo, or based on a pre-existing natural paptide… Thus the origin could be specified in a specific note of the Table. In addition, when the sequence is not provided (possibly as too long for being included in the column, the literature reference.
5- Tables 2 and 4, please specify for LvHemB1 and B11, which regions/domains in the amino acid sequence of L. vannamei hemocyanin are concerned (for instance the copper-containing domain for B11) and lead to these two anticancer peptides; it is worth noting that hemocyanin itself has anticancer properties () and that another region (C-terminus) of hemocyanin (PvHct) leads to an antifungal compound (Petit VW et al. Biochim Biophys Acta. 2016). It should thus be interesting to point the different locations of the different fragments and discuss these aspects in the review.
6- Table 5, reference 136, please specify that GA is gramicidin A, as Aneurinibacillus migilanus (previously Bacillus brevis) also produces gramicidin S, which has a different mechanism of action and presumably a different activity on cancer cells.
Minor points
- In all legends to Figures and Tables the inappropriate and randomly used capital letters have to be suppressed. In addition, the names of peptides are common names, and as such they should not have a capital letter (examples: cecropin lines 68, 104-105, 113, 158…, melittin lines, 129, 132, 136, 138, 139, 141, 310, beclin and bovine lactoferricin line 260, cyclin lines 279-283, etc). In addition, it should be homogeneous throughout the manuscript and not alternatively with and without capital letter.
- Throughout the paper, including Tables and Figures and literature references, the names of genus and species of bacteria and other organisms have to be homogeneously in italics and without capital letter for the species name (as an example, line 348, change “Musca Domestica Cecropin” to “Musca drosophila (ital) cecropin”.
- Paragraph lines 65-69: avoid too repetitive style (“have/has anticancer activity”); please reword.
- Lines 77-78: confusing sentence (“these drugs will kill normal cells except cancer cells”) that could be understood as these commonly used drugs only kill healthy cells and not malignant cells. Please reword.
- Line 99 suppress useless capital letters (Yoshitaka Kamino) to make the manuscript homogeneous.
- In tables, please pay attention to appropriate sizes of the columns and make sure that words are not inappropriately cut, due to a lack of space in the column
- Table 4: change “maize” to “Zea mays” (in italics) to make the table homogeneous; for SK84, change “Drosophila” to “Drosophila virilis” (italics);
- Table 4, for nisin (at the level of reference [93], change “Streptococcus and Lactococcus” to “Streptococcus spp. and Lactococcus spp.” to indicate that several species in the genus are producers.
Author Response
请参阅附件。

Reviewer 2 Report
Comments and Suggestions for Authors
This review article is focused on the potential uses of antimicrobial peptides (AMPs) in gastrointestinal cancer. Naturally occurring AMPs, their derivatives, and de novo designed AMPs have been well investigated for their potential applications to overcome the increasing antibiotic resistance. Some studies have also reported the possibilities for use against cancer. So, this is an excellent topic and timely to publish. The review article needs a careful revision. Since this is a review article, previous studies should be included.
1) The authors have covered topics focused on cancer-related. But, AMPs were originally meant to be for antibiotic applications. This should be mentioned. It is useful to include references to recently published articles and comprehensive review articles in this area:
- Won et al Biophysical Chemistry 24 February 2023.
- - Biochim Biophys Acta. 2009 Aug;1788(8):1680-6.
- - Roversi et al Biophysical Chemistry8 June 2023
- - Halder et al Biophysical Chemistry5 January 2022
- - Mohid et al Biophysical Chemistry22 March 2022
2) The mechanisms of AMPs' action should be briefly mentioned. These mechanisms are thought to be very common against bacteria or cancer cells. So, it is useful to mention even if there are reviews that covered them.
- Biochim Biophys Acta. 1999 Dec 15;1462(1-2):11-28.
- Biochem Cell Biol. 2002;80(5):667-77.
3) Limitations of AMPs and reasons for the unsuccessful studies should be included.
4) LL-37 is one of the most important AMPs. This also has significance against cancer and potential applications are very high. This is nicely mentioned by the authors. But, it would be useful to include the previous studies on this very important human AMP.
- Biochim Biophys Acta. 2006 Sep;1758(9):1408-25.
Comments on the Quality of English Languagefine
Author Response
请参阅附件。
